# PSYCHOPHYSICAL VS. LEARNT TEXTURE REPRESENTATIONS IN NOVELTY DETECTION

## ABSTRACT

Parametric texture models have been applied successfully to synthesize artificial images. Psychophysical studies show that under defined conditions observers are unable to differentiate between model-generated and original natural textures. In industrial applications the reverse case is of interest: a texture analysis system should decide if human observers are able to discriminate between a reference and a novel texture. For example, in case of inspecting decorative surfaces the detection of visible texture anomalies without any prior knowledge is required. Here, we implemented a human-vision-inspired novelty detection approach. Assuming that the features used for texture synthesis are important for human texture perception, we compare psychophysical as well as learnt texture representations based on activations of a pretrained CNN in a novelty detection scenario. Additionally, we introduce a novel objective function to train one-class neural networks for novelty detection and compare the results to standard one-class SVM approaches. Our experiments clearly show the differences between human-vision-inspired texture representations and learnt features in detecting visual anomalies. Based on a digital print inspection scenario we show that psychophysical texture representations are able to outperform CNN-encoded features.

## 1 INTRODUCTION

The idea of describing the appearance of textures using statistics goes back to the early work by Gibson (Beck & Gibson, 1955; Gibson, 1950) and by Julesz (Julesz, 1962; 1981; Julesz et al., 1978). Since then, a number of Markov random field texture models for modelling and characterizing textures using the statistical description of local neighbourhoods were introduced by Cross & Jain (1983) and Geman & Geman (1984). Another category of models tries to find a plausible texture representation for the early visual system of humans (Heeger & Bergen, 1995; Portilla & Simoncelli, 2000; Safranek et al., 1990). These human-vision-inspired models are based on a decomposition of the texture to frequency and orientation bands. The well-known texture model by Portilla & Simoncelli (2000) (PS-model) and the recently published image-computable spatial vision model by Schütt & Wichmann (2017) (SW-model) are two representatives of such psychophysical models. The PS-model is based on joint statistics of complex wavelet coefficients (Simoncelli & Freeman, 1995) and focuses on synthesizing realistic textures. The SW-model is based on a log-Gabor decomposition followed by an accelerating nonlinearity and normalization. In contrast to these psychophysically motivated models (assuming a plausible image representation for the early visual system), Gatys et al. (2015) introduced a texture model based on features of a pretrained deep convolutional neural network (CNN). Using these CNN-encoded features, the model shows impressive results in generating artificial textures (Gatys et al., 2016; Wallis et al., 2017).

Specific texture representations are required in industrial applications, e.g. for pattern recognition tasks or modelling human perception. In this work we focus on representing textures in such a way that visual anomalies can be detected when comparing reference and novel examples. This is particularly required whenever a reference texture should be reproduced visually indistinguishable. In general, the task of identifying data that differs from a reference is known as novelty detection. In contrast to classification tasks, only one class of labelled data (reference texture) is available. As an example of application, we use images of artificial wood textures which we digitised using line-scanner cameras, installed in an industrial digital printer for laminate flooring. Here, the reference texture is only available digitally as a scanned image of the initially produced reference decor.

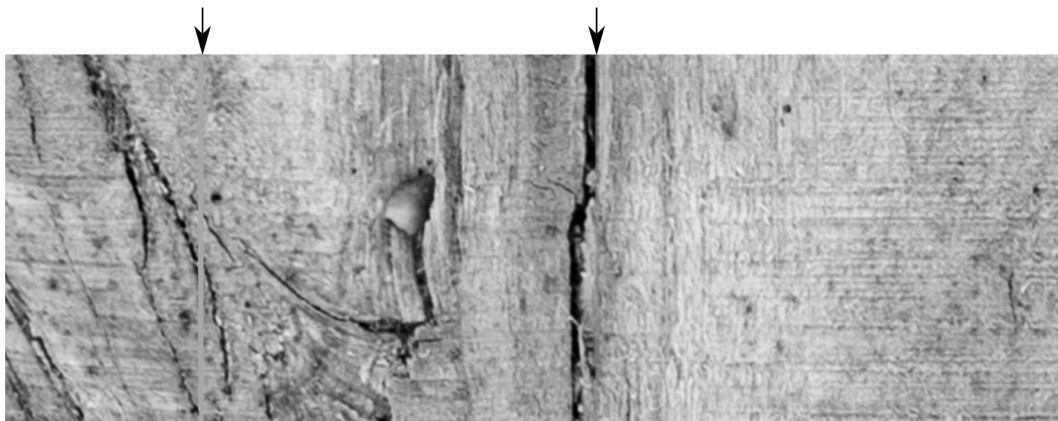

Figure 1: Example of a visible and non-visible line anomalies in an artificial wood texture. The texture was printed in an industrial single-pass print line for laminate flooring. Both line anomalies have the same size, and both are related to the same primary colour.

From a machine learning point of view, there are no labelled training samples available to train a supervised classifier. Another critical point is the interdependency of the surrounding texture and the visibility of specific anomalies. Hence, isolated anomaly consideration is not feasible (cf. Figure 1). In decorative surfaces, e.g., a wallpaper of a printed brick wall or a cupboard with an artificial wood texture, each print should be indistinguishable from any other. Since humans perceive small variations in a texture when comparing two examples, the detection of visual anomalies perceived by a human observer is an open challenge in computer vision and a continuous problem in industrial applications.

To learn the model of a reference texture and detect visual anomalies in novel examples, we present an unsupervised one-class neural network approach using a parametric texture representation. We summarize our main contributions as follows:

- We introduce a novel objective function (MinMax-loss) to train one-class neural networks for novelty detection in textures.
- We introduce psychophysically motivated texture models - PS-model and SW-model - for novelty detection using one-class classifiers and provide a comparison with deep CNN texture features.
- We show texture dependent differences between the models which are of interest in visual surface inspection applications.

## 2 RELATED WORK

There are many approaches for finding outliers in data or detecting them when observed. A standard approach in novelty detection is finding outliers i.e. data points that do not belong to a known group with clustering approaches e.g. Gaussian-Mixture-Models or k-means (Grünauer et al., 2017). This broad range of methods uses distance, density or threshold-measures to decide whether a new data point belongs to an existing cluster or is unknown (Scott & Blanchard, 2009). However, these methods rely on a good similarity or distance measure which is often not available, hard to compute or just not working with high-dimensional feature spaces due to the curse of dimensionality (Steinbach et al., 2003). Besides such classical clustering techniques, support vector machines (SVM) can be used for modeling a single class (Schölkopf et al., 1999). Compared to binary classification one-class SVMs (OC-SVM) divide a single class of reference data into two sets by fitting a hyperplane so that a small subset of normal data is treated as outliers (anomaly data). In addition, the SVM algorithm maximizes the distance between normal data and the subset of outliers. Approaches based on the OC-SVM algorithm have already been successfully applied to detect visual anomalies on textured surfaces and widely exist in industrial contexts (Jahanbin et al., 2009). Beyond SVM approaches there are subspace or latent code techniques which find anomalies by projecting data onto

a chosen subspace and evaluate the reconstruction error (Hoffmann, 2007) or cluster assignments (Xie et al., 2016). Common techniques for projecting data on a subspace are related to principal component analysis (PCA) or autoencoders. One main difficulty of subspace or SVM-methods is computational complexity. The need for matrix inversion or at least pairwise function evaluations, in case of kernel-methods, often is runtime critical (Pimentel et al., 2014). Parametric models such as autoencoders do not suffer from this issue and can be applied to large datasets. However, due to non-linearity, they are harder to optimize. Finally, hybrid approaches try to combine robust PCA (RPCA) with autoencoders and separate noise from the reference data (Zhou & Paffenroth, 2017). RPCA improves reconstruction-based methods, as it removes noisy training examples from the learnt latent representations making it a better fit for the normal data. Our approach follows this idea and introduces a neural network training technique that explicitly learns a hyperplane separating reference data and data with same image statistics.

## 3 NOVELTY DETECTION

For novelty detection on textures without any prior knowledge of anomalies, a reference model is required. Based on this reference model a decision can be made whether or not a novel observation is anomalous. Here, we introduce a novel objective function that we later use to train a model of a given reference texture using a one-class approach based on a neural network.

Our approach is motivated by narrowing the range of predicted reference outputs. This is achieved by minimizing $\max(\hat{\mathbb{Y}}_{\text{ref}}) - \min(\hat{\mathbb{Y}}_{\text{ref}})$ in the output space. To find a non-trivial solution, i.e., $\max(\hat{\mathbb{Y}}_{\text{ref}}) > \min(\hat{\mathbb{Y}}_{\text{ref}})$, a second training input is used for regularization. This input is based on the same image statistics as the reference, that is assured by randomly shuffling the reference texture across image dimensions (cf. Section 4.1). In particular, we separate the maximum predicted value of all shuffled reference inputs from the minimum predicted value of all reference inputs - hence the name *MinMax-loss*. The resulting objective is given by

$$\mathcal{L}_{\text{minmax}} = \big( \max(\hat{\mathbb{Y}}_{\text{ref}}) - \min(\hat{\mathbb{Y}}_{\text{ref}}) \big) + \big( 1 - \tanh \big( \min(\hat{\mathbb{Y}}_{\text{ref}}) - \max(\hat{\mathbb{Y}}_{\text{shuffled}}) \big) \big) + \frac{1}{2} \|\boldsymbol{W}\|^2, \quad (1)$$

where $\max(\hat{\mathbb{Y}}_{\text{shuffled}})$ is the maximum predicted value of all shuffled, and $\min(\hat{\mathbb{Y}}_{\text{ref}})$ is minimum predicted value of all reference inputs, and $\boldsymbol{W}$ are the model weights.

In contrast to hinge loss based one-class approaches, such as one-class SVMs (cf. Schölkopf et al. (1999)), that use a signum function as decision function, we use an interval-based decision function. Since we minimize the range of the predicted reference output distribution, our decision function becomes

$$f_{\text{dec}}(\hat{y}, \hat{Y}_{\text{ref}}) = \begin{cases} 1, & \text{iff } Q_\nu(\hat{\mathbb{Y}}_{\text{ref}}) < \hat{y} < Q_{1-\nu}(\hat{\mathbb{Y}}_{\text{ref}}) \\ 0, & \text{otherwise.} \end{cases} \quad (2)$$

Here, $\hat{y}$ is the predicted output to be classified, $Q_\nu(\hat{\mathbb{Y}}_{ref})$ is the $\nu^{th}$ quantile of all predicted reference values $\hat{\mathbb{Y}}_{ref}$, and $\nu$ is the tolerated fraction of reference examples being classified as anomalous. Given a new input example, our decision function yields 1 if the prediction is within the reference interval and 0 otherwise. Hence, anomalies are labelled 0.

The previously introduced MinMax-loss is used in a neural network based one-class classifier to model the reference textures and detect anomalies in novel observations. We use a sliding window approach to process textures independently from the input dimensions. Since our MinMax-loss depends on $\hat{\mathbb{Y}}_{\text{ref}}$ and $\hat{\mathbb{Y}}_{\text{shuffled}}$ the classifier is trained by alternately propagating reference and shuffled texture patches through the same network (cf. Section 4.1).

### 3.1 PS-MODEL BASED FEATURES

Studies in the field of visual psychophysics (e.g. Wallis et al. (2017)) were concentrated on the discriminability of textures synthesized with the PS-model from the original when being presented to human observers. The results show that depending on the type of texture and the viewing conditions, the generated images are indistinguishable from the original image. With the goal of using a representation that captures the appearance of a texture, we use the PS-model as a part of our preprocessing.

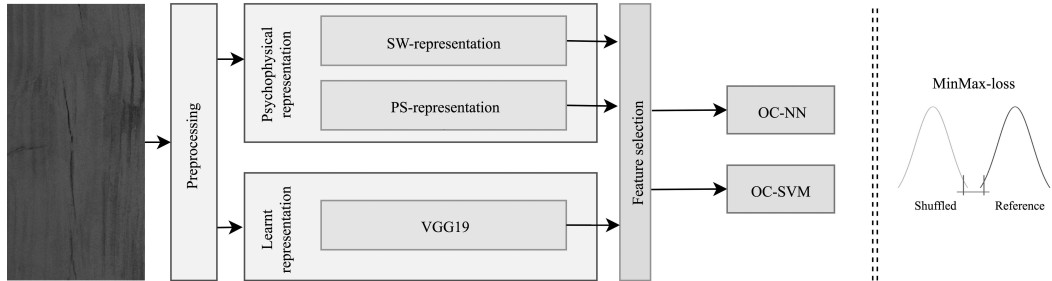

Figure 2: Illustration of the processing pipeline: First, the input image is being preprocessed. Afterwards, the particular representations are being computed. Finally, the representations are being evaluated by both one-class classifiers and compared among each other. On the right-hand side the behaviour of our MinMax-loss, i.e. separating the output distributions of the reference and the shuffled texture, is illustrated.

The decomposition is based on an overcomplete complex wavelet transformation. Therefore, a directional, non-uniform filter bank with cascaded filters is applied. In this context, non-uniformity means that filter responses have different bandwidths. First, the image is decomposed by using high- and low-pass filters. The sub-sampled low-pass response is filtered iteratively with the same set of filters. The number of iterations corresponds to the amount of scales ($N_S$) and the number of directional filters per iteration corresponds to the amount of orientation sub-bands ($N_O$) of one scale (Simoncelli & Freeman, 1995).

In Portilla & Simoncelli (2000), the parametric texture model that is able to synthesize artificial and natural textures is introduced. The model is based on joint statistics of the complex wavelet coefficients obtained from the preceding decomposition. The resulting parameters contain image statistics such as mean, variance, skewness, and kurtosis of the input image as well as the single filter responses. For feature preprocessing, we use four spatial orientations and choose a spatial neighbourhood of $9 \times 9$. Depending on the size of the sliding window and considering that the dimension has to be a multiple of $2^{(N_S+2)}$, we use up to four scales.

## 3.2 SW-MODEL BASED FEATURES

The model by Schütt & Wichmann (2017) is a psychophysical, image-computable model for early spatial visual processing. The model is based on an image preprocessing, a decomposition into spatial frequencies and orientations followed by a nonlinearity and normalization. As a part of the preprocessing, contrast images are converted to luminance images. Furthermore, optical distortions (eye optics, contrast sensitivity function and cut-out of the fovea) are applied. For decomposing the image into spatial frequencies and orientation channels, a complex log-Gabor filter bank with $8 \times 12$ filters is used. Finally, the nonlinearity and normalization are applied to each channel response.

Our parameter values are based on the work of Schütt & Wichmann (2017). We use the following configuration: Gabor standard deviations $\sigma_F = 0.5945$ for the spatial frequency and $\sigma_\theta = 0.2965$ for the orientation, nonlinearity constant $C = 0.0027$, nonlinearity exponent $p = 2.1698$, difference exponents $q = 1.8667$, and pool orientation $\omega_\theta = 0.1112$.

The SW-features are 96 times overcomplete. To limit the input size of classifiers, we must use a reduced version of the SW-representation. Therefore, we evaluated different averaging methods - maximum and mean activity per channel after normalization $x_{\text{SW}}^{\text{apc}}$, maximum and mean filter responses $x_{\text{SW}}^{\text{fr}}$ after normalization as well as combined versions. Note, the number of features within the resulting descriptor $x_{\text{SW}}^{\text{apc}}$ are independent of the input size. Whereas the descriptor $x_{\text{SW}}^{\text{fr}}$ is of size $2 \times m \times n$ , where $m$ and $n$ are the height and width of the input image.

### 3.3 LEARNT TEXTURE REPRESENTATION

In contrast to handcrafted psychophysical models of human-vision, learnt texture representations are based on the activations of a neural network. Additionally, it is important to note that unlike autoencoder approaches, the network is not explicitly trained to represent a particular texture, but e.g., optimized for a classification task. In this work we use a VGG-19 network introduced by Simonyan & Zisserman (2014), which was pretrained on ImageNet (Russakovsky et al., 2015). Inspired by the work of Gatys et al. (2015) we use the *feature maps* of an arbitrary pretrained convolutional layer to compute so-called *normalized features*. In particular, the normalized features are represented by a vector containing the feature maps of a certain layer which are normalized by the squared $\ell_2$-norm. The resulting vector becomes $g_i^l = \|\boldsymbol{f}_i^l\|^2$, where $\boldsymbol{f}_i^l$ is the $i^{\text{th}}$ vectorized feature map in the output of layer $l$. Like the *Gramian features* from Gatys et al. (2015), this representation has the advantage of being independent of the input size.

## 4 EXPERIMENTS

Inspired by a real world application, we conducted a broad range of novelty detection experiments to (1) compare the human-vision-inspired PS and SW texture representations with the CNN-encoded features, and to (2) evaluate the performance of our one-class neural network (cf. Figure 2). In the following, we briefly summarize our experiments and continue to detail meaningful results.

As mentioned before, we tested different averaging methods to reduce the size of the SW-feature vector. Based on the evaluation results across all experiments, we use a concatenated SW-feature vector $\boldsymbol{x}_{\text{SW}}^{\text{mean}} = \{\boldsymbol{x}_{\text{SW}}^{\text{fr}}, \boldsymbol{x}_{\text{SW}}^{\text{apc}}\}$ for reporting our results. Additionally, we conducted experiments where the PS-features $\boldsymbol{x}_{\text{PS}}$ were tested with and without image statistics. Experiments showed that these image statistics impact detection results, therefore we evaluate both. Furthermore, we analysed the performance of our normalized features based on the feature maps of different VGG-19 layers, i.e. *conv1_1*, *pool1*, *pool2*, *pool3*, *pool4*, and *pool5*. Best results were achieved by using the feature maps of layer *pool4*. Therefore, the results reported in the following are based on normalized features computed from the feature maps of *pool4*.

### 4.1 EXPERIMENTAL SETUP

Our approach was tested in a surface inspection system, which is installed in a digital print line for artificial wood decors. In this appliance it was possible to evaluate 74 different printed decors. For reporting our results we chose two representative examples - a pseudo-periodic brick wall texture (*red-bricks*) as well as a typical wood texture (*C7-G-10-2*), see Figure 3. In order to provide reproducible anomalies for further investigations, we report our results using synthesized anomalies similar to the most common defects in single-pass digital printing, the so-called *nozzle faults* or *line defects*. These failures occur randomly and the human ability to perceive line defects heavily depends on the surrounding texture (cf. Figure 1).

All experiments use a three-layer neural network (OC-NN) with a single linear output neuron. As a result of previous experiments, we chose the same number of hidden neurons as input neurons and use a sigmoid activation function. Weights are optimized using our MinMax-loss (cf. Equation 1). We use gradient descent optimization with a learning rate of $0.001$. All models are trained for 1000 epochs in an off-line scenario to ensure convergence.

We ran all experiments on 8-bit contrast images, which we normalize to $[0, 1]$. For tests with RGB textures, we first convert the images into greyscale using the weighted sum $I = 0.2989R + 0.587G + 0.114B$. We use a sliding window $s$ of size $s_{\text{size}}$ 64 px, 128 px, and 256 px and a stride $s_{\text{stride}}$ of size $s_{\text{size}}/2$ for processing texture images size-independently. Since the VGG-19 network was trained on RGB images and expects three-channel input, we duplicated our greyscale images into three channels. We implemented our OC-NN, the VGG-19 network, the SW-model, and the PS-model in *Python* (3.6.5) using *PyTorch* (0.4.1), *Torchvision* (0.2.1), *Imageio* (2.3.0), *NumPy* (1.14.3), and *SciPy* (1.0.0). For the OC-SVM comparisons, we used the implementation from *scikit-learn* (0.19.1). We conducted all experiments using *linear*, *sigmoid*, and *RBF* kernels. The default $\gamma$-value, i.e. $\frac{1}{\text{number of features}}$, was used for the sigmoid and RBF kernels, since other values did not improve the results. For all OC-SVM experiments, we choose $\nu = 0.001$ as the upper bound on the fraction of training errors. For the remaining parameters, we took the *scikit-learn* default parameters.

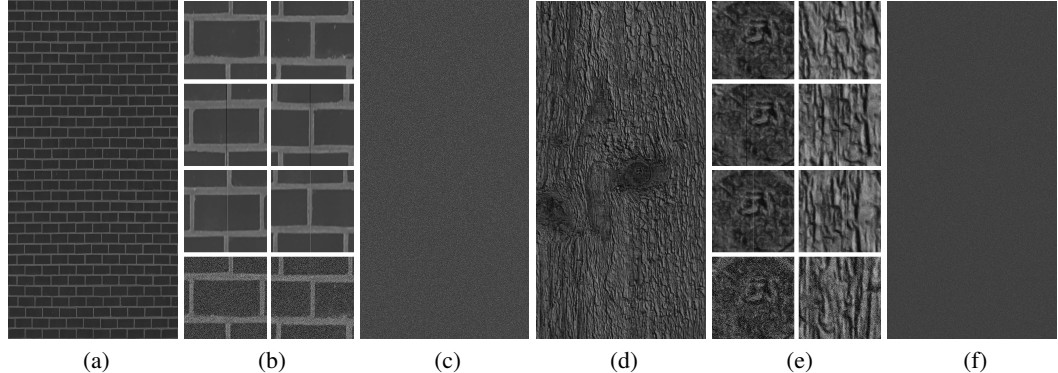

|     |     |     |     |     |     |
| --- | --- | --- | --- | --- | --- |
| (a) | (b) | (c) | (d) | (e) | (f) |

Figure 3: Illustration of (a) an excerpt of the *red-bricks* (1024 × 1024 px) texture, (b) examples for reference, line anomaly, and noise anomaly patches, and (c) an excerpt of the shuffled reference texture used for training our OC-NN. Analogously to that, (d) to (f) illustrate the aforementioned details for the texture *C7-G-10-2* (2048 × 4096 px).

## 4.2 MODEL EVALUATION

All models are trained in a novelty detection scenario, where only a single example of the reference texture is available. As mentioned before, to provide reproducible anomalies for further investigations, we used synthesized anomalies similar to the most common line defects in digital printing (cf. Section 4.1). Therefore, we add and subtract an offset value from the pixels with different intensities (i.e. ±128, and ±255) and optionally extend the anomaly to adjacent lines (i.e. 1, 2, 4, and 8). All models are tested against the same 100 randomly placed anomalies. Models are evaluated using the area under the ROC curve (AUC) measure (we also evaluated average precision-recall with same results throughout all experiments). Again, for all models we choose $\nu = 0.001$ as upper bound on the fraction of training errors.

## 4.3 PERFORMANCE COMPARISON OF PARAMETRIC TEXTURE MODELS

To compare the performance of the different texture representations, we train our OC-NN (cf. Section 3) as well as an OC-SVM (cf. Section 4.1) on the particular texture features. The one-class classifiers are provided with the texture representations resulting from the particular preprocessing step without any normalization or whitening. Here, we report our results based on the reference textures introduced in Section 4.1. The results achieved by using different anomaly sizes and intensities are shown in Figure 4. On the pseudo-periodic *red-bricks* texture all texture representations achieve an AUC greater than 0.5, when using our OC-NN classifier. In addition, from the novelty detection point of view and without any consideration of the visibility of anomalies, on *red-bricks* the best AUC is achieved by using the CNN-encoded features to train our OC-NN model (cf. Figure 4a). As shown in Figure 4c, the OC-NN model trained with SW-features has the overall best AUC for small anomaly sizes, but at the same time, the AUC decreases for larger anomaly. As illustrated in figures 4b and 4d, OC-SVM models trained with the SW-representation as well as the CNN-encoded features are not able to detect anomalies. Using the PS-representations, the OC-SVM is able to detect anomalies on the pseudo-periodic *red-bricks* texture as well as on the *C7-G-10-2* texture.

## 4.4 COMPARISON OF NOVELTY DETECTION PERFORMANCE WITH ADDITIVE NOISE

When inspecting printed textures for decorative surfaces, typical sources of noise are related to the printing process, the environmental conditions, or the digitalisation process. From the human-vision-inspired novelty detection point of view, noise must be detected as anomaly if and only if it is perceived by a human observer. Therefore, we evaluated the models' behaviour when distorting the input image with different types of noise, such as Laplacian or Gaussian noise. For reporting our results we train our OC-NN model for both reference textures and evaluate the performance of the model using Gaussian noise distorted inputs. As shown in Table 1, the OC-NN model trained

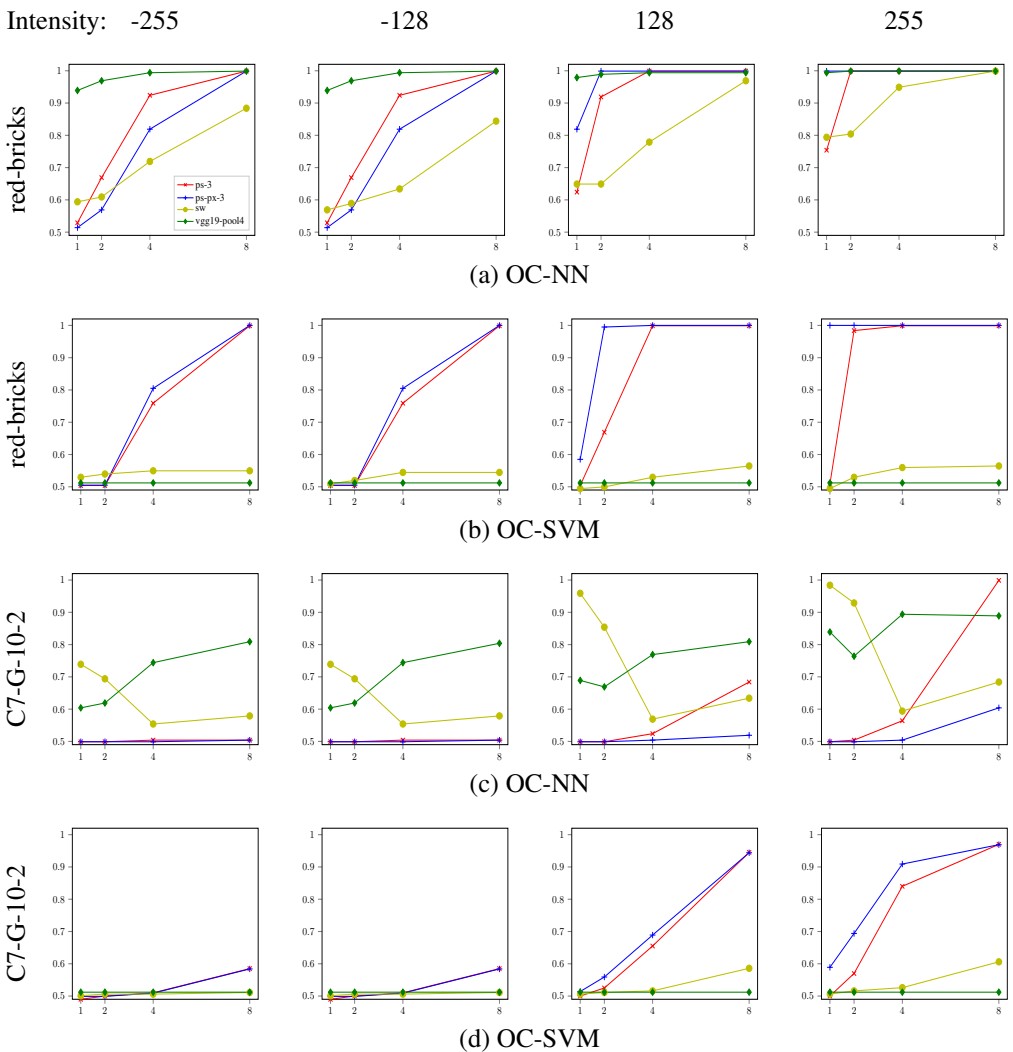

Figure 4: Comparison of classifier performances on the previously introduced representations of the textures *C7-G-10-2* and *red-bricks*. The resulting AUC scores are plotted on the y-axis and the width of the line anomalies on the x-axis. Each particular column represents the different anomaly intensities ($\pm 128$ and $\pm 255$) used. Furthermore, each row illustrates the results of a particular classifier.

with PS-features is not able to detect noise. The model trained with the VGG-19 texture representation achieves AUC scores greater than $0.9$ on both textures, when evaluating examples distorted by additive Gaussian noise with a standard deviation of $\sigma = 0.1$.

Table 1: Comparison of novelty detection performances (AUC scores), when evaluating examples distorted by additive Gaussian noise. The particular texture representations were evaluated on both reference textures with our OC-NN model. AUC scores higher than $0.80$ are highlighted with boldface digits.

| Texture | C7-G-10-2 | | | red-bricks | | |
|---|---|---|---|---|---|---|
| representation | $\sigma = 0.1$ | $\sigma = 0.01$ | $\sigma = 0.001$ | $\sigma = 0.1$ | $\sigma = 0.01$ | $\sigma = 0.001$ |
| VGG | **0.94** | 0.58 | 0.50 | **0.99** | **0.85** | 0.50 |
| SW | 0.59 | 0.58 | 0.50 | **0.99** | **0.98** | 0.50 |
| PS | 0.50 | 0.50 | 0.50 | 0.50 | 0.50 | 0.50 |

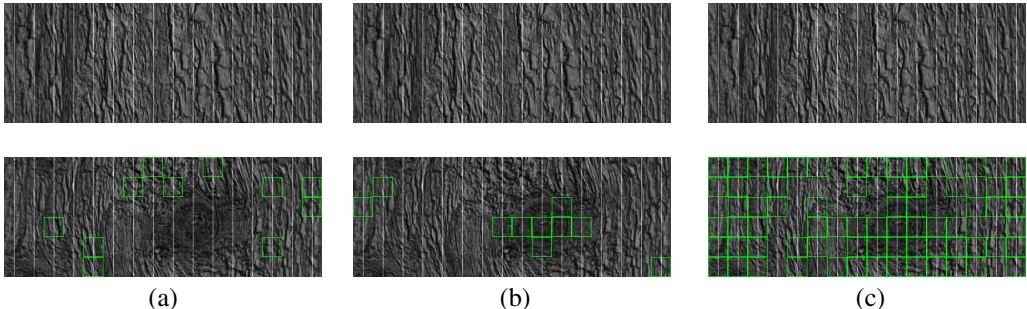

(a)  (b)  (c)

Figure 5: Visualisation of detected line defects of the particular texture representations, i.e. (a) PS-representation, (b) SW-representation, and (c) learnt texture representation. The line defects were applied to the texture *C7-G-10-2* with an intensity of $+128$ and width of $4$ px. Line defects that were detected are highlighted by a surrounding (green) square.

### 4.5 ANALYSING DIFFERENCES IN FEATURES (MODEL COMPREHENSION)

The differences of the representations become more clear when comparing the results visually. Figure 5 shows line anomalies (4 px in width and an intensity of 128) on the *C7-G-10-2* texture. For evaluation purposes anomalies are not placed randomly, but everywhere on the texture with no overlapping. The huge difference between psychophysical and learnt representation becomes obvious when looking at the highlighted detected anomalies. While the learnt texture representations detect almost any anomaly, psychophysical are more selective. The classifier trained with the SW-representation preferably detects anomalies on darker patches, while the classifier trained with the PS-representation seems to detect anomalies in an arbitrary manner.

## 5 CONCLUSION

In this paper, we evaluated the performance of novelty detection in digital print inspection using psychophysical and learnt texture representations. First we introduced state-of-the art methods for statistical and early vision based modelling of textures. While the model by Portilla & Simoncelli (2000) focuses on synthesizing realistic textures, the model by Schütt & Wichmann (2017) focuses on modelling the human early vision system in an image-computable way. Another branch of texture modelling uses learnt representations based on features of a pretrained CNN (Gatys et al., 2015). Based on the aforementioned features we learnt the reference model of a texture and detected visual anomalies in novel examples. Therefore, we introduced a novel objective for training neural network based one-class classifiers for novelty detection (OC-NN). Additionally, we compared our OC-NN approach with an OC-SVM based classifier and showed superior results in our application scenario.

All texture representations achieve reasonable results, when being evaluated with our OC-NN approach on a quasi-periodic texture (cf. *red-bricks*). However, when being evaluated by an OC-SVM based classifier, anomalies cannot be detected using SW- as well as VGG-19-features. This might be due to the lack of preprocessing, such as z-normalisation or whitening, but this is referred to future work. Altogether the learnt texture representations provide a set of features for detecting novelties, that performs well, whether or not they are perceived by a human observer. Furthermore, PS-features provide a texture-dependent descriptor, that achieves reasonable results on quasi-periodic textures. Finally the SW-features outperform the PS- and VGG-19-features for small anomalies on aperiodic textures, which is great from the application point of view. However, when anomalies dominate (fixed patch size and increasing anomaly size), the detection rate decreases.

As a part of the work for this paper, we evaluated our one-class model for novelty detection on a broad range of industrial printed decors, but more work using psychophysical data from experiments is needed. We plan to validate selected anomalies in psychophysical experiments. Further work will also include evaluating different pooling and averaging methods for SW-features. Finally, we plan to fine-tune an OC-SVM for being able to use SW-features.

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
