# OpenReview forum: "Psychophysical vs. learnt texture representations in novelty detection"
_ICLR.cc/2019/Conference_

### Official Review · AnonReviewer1 · 2018-10-30
**Psychophysical vs. learnt texture representations in novelty detection**

**Rating:** 1
**Confidence:** 3

**Review:**



The authors describe an anomaly/novelty detection method based on handcrafted features + VGG based features.

I think the paper is out of the scope of the conference (the only part dealing with learned representations uses VG), plus it addresses a problem whose relevance is not correctly motivated. Finally, the method is quite basic, and is not compared to any state of the art method for novelty detectiobn.

In "... the detection of visual anomalies perceived by human observer is an open challenge… " can you provide references of people working in this particular problem?

The review of related work seems obsolete, can you provide more recent references (in addition to "historical" ones). More importantly, please provide references of anomaly detection from textures

---

### Official Review · AnonReviewer2 · 2018-11-03
**Interesting topic but insufficient analysis and evaluation**

**Rating:** 3
**Confidence:** 4

**Review:**

The submission investigates the problem of detecting perceptual anomalies in visual textures.
It proposes features from three different models, the Portilla & Simoncelli texture model (PS), the Spatial Vision model by Schuett and Wichmann (SW) and CNN features from the VGG network. From these features it trains two anomaly detectors: one out of the box one-class SVM and a 3 layer neural network. The network is optimised with a loss function that encourages output values for the original texture to be larger than for a white-noise image obtained by shuffling its pixels. At the same time the range of output values for the original texture is encouraged to be small.

The performance of the different approaches is evaluated using synthetic anomalies. However no distinction between perceptually striking and perceptually negligible anomalies is made and quantitative results are only reported for all synthetically generated anomalies.
Two attempts are made to control if an approach specifically picks up on perceptually striking anomalies.
a) detection rate on gaussian noise as a proxy for perceptually negligible anomalies
b) anecdotal evidence from visual inspection.

I do not think that either of the two controls is sufficient to make a clear statement about which method is best in detecting perceptually striking anomalies. Therefore my main concern is that the performance evaluation is not suitable to achieve meaningful results.

Furthermore the technical depth of the submission appears fairly limited. The main original contribution is the CNN loss that is introduced. However, the loss does not strike me as particularly compelling. It resembles a classifier between textures and white noise samples with the same pixel-wise statistics. I am not sure why this should be particularly suited to detect perceptual anomalies.

Finally, showing quantitative results from only two textures does not feel like a very comprehensive analysis.

In general  the submission tackles an interesting research topic. However, to show meaningful results I believe that one has to collect psychophysical data for the anomalies of interest to distinguish between anomalies that are perceptually detectable and those that are not.
With such a test set one could then start testing hypothesis on which feature representation is most appropriate to model the perceptual results or optimise features directly to match human psychophysical results (similar to the study by Berardino et al. 2017 [1]). In its current form I am not sure what I can learn from the submission both in terms of anomaly detection and feature spaces particularly suited to detect perceptual anomalies in visual textures.

[1] Eigen-distortions of hierarchical representations
A Berardino, V Laparra, J Ballé, E Simoncelli
Advances in neural information processing systems, 3530-3539

---

### Official Review · AnonReviewer4 · 2018-11-12
**Experimentally Limited**

**Rating:** 3
**Confidence:** 3

**Review:**

This paper considers detecting anomalies in textures. For this task they use VGG-19 features and two human-inspired features from Portilla & Simoncelli and Schutt & Wichmann.
With these features, they train one-class anomaly detectors. One such anomaly detector is a one-class SVM, and they introduce a loss for one-class neural networks.

The novelty in this paper comes from the problem setup which I have not seen treated before. The loss function they propose also appears original.

However, comparisons are limited. They compare against OC-SVMs, but these are known to be weaker than several types of anomaly detectors [1]. This paper would also do well to ground itself in more recent research on deep anomaly detection [2]. Likewise, the problem setting is limited. In all, experimentation could use more breadth and depth.

[1] Andrew F. Emmott, Shubhomoy Das, Thomas Dietterich, Alan Fern, Weng-Keen Wong. Systematic Construction of Anomaly Detection Benchmarks from Real Data. ODD, 2013.
[2] Dan Hendrycks and Kevin Gimpel. A Baseline for Detecting Misclassified and Out-of-Distribution Examples in Neural Networks. ICLR, 2017.

---

### Official Review · AnonReviewer3 · 2018-11-28
**Interesting task but lack of novelty**

**Rating:** 3
**Confidence:** 3

**Review:**

This paper focuses on novelty detection and shows that psychophysical representations can outperform VGG-encoder features in some part of this task.

Novelty:
It is the first time I have seen this novelty detection task. This task could be part of the novelty of the paper. Another novelty comes from the new objective function they introduce.

Weakness:
1. The motivation of the new objective function is not clear to me. It seems that they first design the objective function and then build the interval-based decision function. There is not much intuition given.
2. The experiment lacks of real data. Synthesized anomalies never exists in application. If it is a paper about application, real data is needed.
3. The baseline is too simple. CNN could definitely beat SVM in image classification. Also using extracted features could be better than directly performing SVM on pixels.
4. Do not see the results of OC-SVM in Table 1 even though they say they beat it.
5. Also, I do not see any other reference work for this novelty detection problem. If it is a new problem, a clear definition of the problem is needed. If it is not, more references are needed.

The writing of the paper is clear and easy to understand. But based on all the weakness above and lack of novelty, I think the paper should be rejected for now.

---

### Meta-Review · Area_Chair1 · 2018-12-17
**Interesting question but insufficient analysis**

**Confidence:** 5
**Recommendation:** Reject

**Metareview:**

This paper focuses on the problem of detecting visual anomalies within textures. For that purpose, the authors consider several parametric texture models and train anomaly detection models on the corresponding outputs.

Reviewers were generally positive about the topic under study, but were unanimous in signaling a severe weaknesses in the experimental setup. In particular, in R2 words, "my main concern is that the performance evaluation is not suitable to achieve meaningful results", and "showing quantitative results from only two textures does not feel like a very comprehensive analysis". Moreover, the authors did not respond to reviewers feedback. Therefore, the AC recommends rejection at this time.